# Reinforcement learning with a network of spiking agents

**Sneha Aenugu, Abhishek Sharma, Sasikiran Yelamarthi, Hananel Hazan**
**Philip S. Thomas, Robert Kozma**
University of Massachusetts Amherst
{saenugu, asharma, syelamarthi, hhazan, pthomas, rkozma}@cs.umass.edu

## Abstract

Neuroscientific theory suggests that dopaminergic neurons broadcast global reward prediction errors to large areas of the brain influencing the synaptic plasticity of the neurons in those regions Schultz et al.. We build on this theory to propose a multi-agent learning framework with spiking neurons in the generalized linear model (GLM) formulation as agents, to solve reinforcement learning (RL) tasks. We show that a network of GLM spiking agents connected in a hierarchical fashion, where each spiking agent modulates its firing policy based on local information and a global prediction error, can learn complex action representations to solve RL tasks. We further show how leveraging principles of modularity and population coding inspired from the brain can help reduce variance in the learning updates making it a viable optimization technique.

## 1 Introduction

Most reinforcement learning algorithms (RL) primarily fall into one of the two categories, value function based and policy based algorithms. The former category of algorithms such as Qlearning Watkins and Dayan (1992) express the value function as a mapping from a state space to a real value, which indicates how good it is for an agent to be in a state. The latter class of algorithms such as REINFORCE Williams (1992) learn policy as a mapping from state space to action space, which tells the agent the best action in each state to maximize its reward. Both types of algorithms, thus, need to address the problem of learning a representation of the state space to solve the task at hand. As the task gets more complex, so does the representation to be learned.

Deep networks trained with backpropagation Rumelhart et al. (1986) have a potential for rich feature representations afforded by the hierarchical structure of these networks Mnih and et.al (2015). Nevertheless these methods face certain key challenges (power consumption, inference speed, robustness to adversarial attacks, online learning, etc) which could potentially be addressed by revisiting the computational models of learning and decision making in the brain. Similar to the deep networks, the brain is composed of multiple layers from the neurons that represent stimulus to the neurons that encode the action. But unlike in deep networks, backpropagation is not considered a viable optimization technique in the brain Crick (1989). The propagation of error signals backward from the upstream neurons in backpropagation is deemed biologically implausible. Instead reinforcement learning models, supported by the evidence of neural coding of reward prediction errors in the brain Schultz and Dickinson (2000), provides a plausible theory of learning by facilitating local learning rules.

In this study, we demonstrate that a multi-agent RL framework with each agent modeled after the GLM model of a spiking neuron Pillow et al. (2008), can learn complex stimulus-action mappings with local learning rules and a global high level feedback. The policy of the spiking agent is defined as its firing probability conditioned on the stimulus and spiking activity of other agents. Each

Workshop on Real Neurons and Hidden Units (NeurIPS 2019), Vancouver, Canada.

agent updates its firing policy by descending its local policy gradient modulated by a global reward prediction error. A heirarchical structure is imposed on the network of spiking agents to enable rich feature representations similar to the deep networks. We further explore architectural techniques inspired from the brain such as modularity and population coding that can overcome the challenges in convergence posed by local learning rules.

## 2 Reinforcement learning with a network of spiking agents

### 2.1 Notation

An RL domain expressed as a Markov Decision Process (MDP) is defined by the state space, $\mathcal{S}$, the action space $\mathcal{A}$, a state transition matrix, $\mathcal{P} : \mathcal{S} \times \mathcal{A} \rightarrow \mathcal{S}$ and a reward function, $\mathcal{R} : \mathcal{S} \times \mathcal{A} \rightarrow \mathbb{R}$. A policy is a distribution of action probabilities conditioned on state space and is defined as $\pi(s, a, \theta) = \Pr(A_t = a | S_t = s)$ where $\theta$ denotes the parameters of the policy. $J = \sum_{t=0}^{\infty} \gamma^t R_t$ is the discounted return.

### 2.2 Agent and architecture

Consider a GLM spiking agent shown in the Figure 1. The agent's instantaneous conditional probability of spiking at any time instant $t$, denoted by $\lambda(t)$ depends on the input $\mathbf{x}$, its own spiking history $\mathbf{y}$, the spike train histories of other agents at time $t$, $\{\mathbf{y_i}\}$, through the parameters $\theta = (\mathbf{k}, \mathbf{h}, \mathbf{l}, \mu)$ as

$$\lambda(t) = \sigma(\mathbf{k} \cdot \mathbf{x} + \mathbf{h} \cdot \mathbf{y} + \sum_i \mathbf{l} \cdot \mathbf{y_i} + \mu) \tag{1}$$

where $\sigma$ is the sigmoidal non-linearity, $\mu$ is the bias and $\mathbf{k}, \mathbf{h}, \mathbf{l}$ are stimulus, post-spike and coupling filters as shown in Figure 1. The agent produces a spike at any instant with a probability $p(t) = \lambda(t)$ and remains silent with a probability $p(t) = 1 - \lambda(t)$. Given a spike train of length $k$, the probability of producing a spike train is the product of probabilities of generating independent spikes, $\prod_{t \in t_s} \lambda^{(\tau)}(t) \prod_{t \in t_{ns}} (1 - \lambda^{(\tau)}(t))$, where $t_s$ denotes times at which spike occurs and $t_{ns}$ denotes the times at which there is no spike. At each instant $\tau$ in the MDP scale, the action of the agent is a spike train response of length $k$. The policy of the agent is then expressed as the probability of producing a given spike train $\mathbf{A}_\tau$ in a given state $\mathbf{S}_\tau = (\mathbf{x}, \mathbf{y}, \{\mathbf{y_i}\})$ as shown below.

$$\pi(\mathbf{S}_\tau, \mathbf{A}_\tau, \theta) = \prod_{t \in t_s} \lambda^{(\tau)}(t) \prod_{t \in t_{ns}} (1 - \lambda^{(\tau)}(t)) \tag{2}$$

Thus the policy of each agent represents a mapping from the state space to its action space through its parameters. We now consider a hierarchical network of GLM spiking agents where the first layer of agents receive the stimulus from the state space of the MDP and generate the spike train response as an action at a given time $\tau$. The action space of the first layer of agents becomes the state space of the succeeding layer of agents. The action space of the final layer of the agents is the action space of the MDP. We now describe how a network of spiking agents learn to represent a complex mapping from state space to the action space of the MDP to solve the RL task.

### 2.3 Learning updates

We derive our learning framework based on Thomas and Barto (2011), which theoretically proves that in a network of modular agents describing a policy, descending the policy gradient on the network as a whole is equivalent to descending the policy gradient on each of the agents separately. We follow the formulation of a modular actor critic described in Thomas (2011), where each of the agents in the network receives the reward prediction error from a global temporal difference (TD) critic. For a given agent, the gradient of the expected discounted return $\nabla J(\theta)$ Sutton and Barto (1998) at time $\tau$ can be expressed as

$$\nabla J(\theta) \propto \mathbb{E}_\pi \left[ \delta_\tau \frac{\partial \log \pi(S_\tau, A_\tau, \theta)}{\partial \theta} \bigg| \theta \right] \tag{3}$$

where $\delta_\tau$ is the TD error delivered by the critic and $\pi$ is the policy of the agent given by Equation (2), where $S_\tau$ is the state of the agent and $A_\tau$ is the action/ spike train response at time $\tau$. A

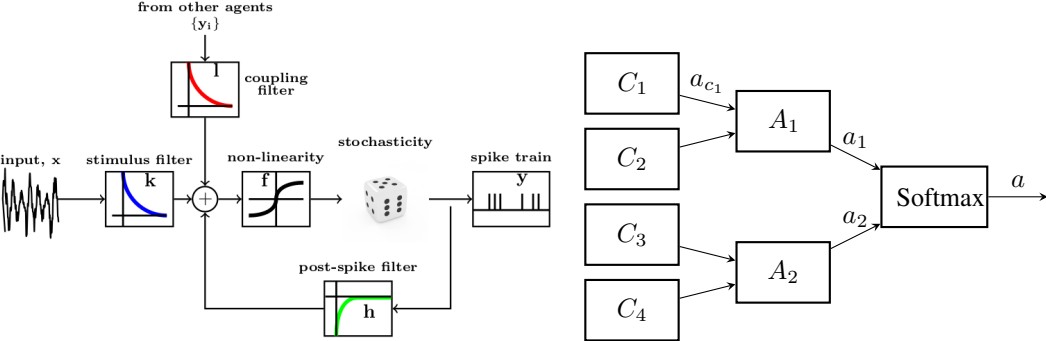

(a) A GLM spiking neuron. A stochastic model of spiking activity with a stimulus filter $\mathbf{k}$ acting on external stimulus $(\mathbf{x})$, a post-spike filter $\mathbf{h}$ accounting for spiking history, coupling filter $\mathbf{h}$ coupling the effect of spiking activity of i other agents $\{\mathbf{y_i}\}$.

(b) In the architecture shown above, the agent $C_1$ affects the spiking of just the agent $A_1$ in the next layer, in contrast with a fully connected architecture, resulting in a modular structure with sparse connections between the layers.

Figure 1

gradient ascent in policy parameters is done to maximize the expected discounted return as follows, $\theta = \theta + \alpha \delta_\tau \nabla J(\theta)$. A stochastic gradient ascent can be performed as $\theta = \theta + \alpha \delta_\tau \frac{\partial \ln \pi(S_\tau, A_\tau, \theta)}{\partial \theta}$.

The log derivative of the policy can be written as

$$\frac{\partial \log \pi(S_\tau, A_\tau, \theta)}{\partial \theta} = \mathbf{x}\Big( \sum_{t \in t_s} (1 - \lambda^{(\tau)}(t)) + \sum_{t \in t_{ns}} (-\lambda^{(\tau)}(t)) \Big) \qquad (4)$$

By the updates $\theta = \theta + \alpha \delta_\tau \frac{\partial \ln \pi(S_\tau, A_\tau, \theta)}{\partial \theta}$, we increase the probability of producing the spike trains which result in a positive TD error and decrease the probability of those that result in a negative TD error. These updates take into account only the local information of the agent and receive no feedback from the upstream agents regarding their contribution to the global action selection, unlike in backpropagation. Hence these updates tend to exhibit high variance as the agent is unaware of the correlation of its spiking policy to the global critic feedback.

## 2.4   Variance reduction

The structural credit assignment problem in deep hierarchical networks is to identify the amount of blame to assign to a particular agent for an error in action selection. One way to reduce the variance in a local learning setup is to introduce a global high level feedback from the network regarding the contribution of the agent to the action selection. However in this study, we focus on the variance reduction by solely targeting architectural design.

**Variance reduction through a modular connectionist architecture**: Brain networks have been demonstrated to have the property of hierarchical modularity, i.e, each module being composed of sub-modules which are in turn composed of several sub-modules. This modular structure is claimed to be responsible for faster adaptation and evolution of the system with changing stimulus conditions Meunier et al. (2010). Jacobs et al. (1991) showed that incorporation of a modular architecture in neural networks results in faster learning compared to a fully connected architecture by decomposing the task into many functionally independent tasks. In this study we demonstrate that such a modular architecture is conducive to local learning rules.

Figure 2 shows a modular connectionist architecture with sparse modular connections instead of a fully-connected architecture. In this network the spiking of an agent in a layer affects only few of the agents in the succeeding layer, thus enabling us to identify and update the agents that are responsible in case an erroneous action is chosen, thereby reducing the variance.

**Variance reduction through population coding**: In the primary motor cortex, it was demonstrated that the decision of the movement is generated by a population of neurons by a weighted vector addition of preferred directions of individual neurons Georgopoulos et al. (1986). This population

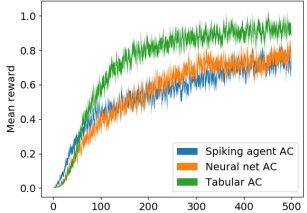
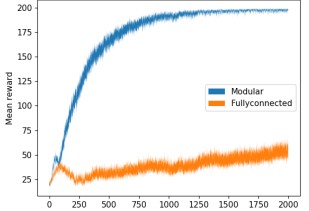
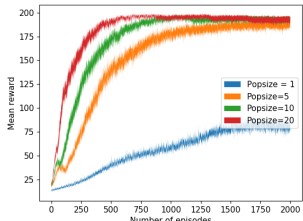

(a) Performance in the gridworld task with a modular spiking agent actor-critic (AC) framework (10 networks) against the baselines of neural network AC trained with backpropagation and tabular AC.

(b) Performance improvement of modular connectionist architecture from Figure 3 against a fully connected architecture in the cartpole task. A population of 10 networks is used in both architectures.

(c) Effect of population coding on the performance in the cartpole task. Increase in population size increases the speed of convergence to the optimum. Modular architecture is incorporated in all these cases.

Figure 2: Learning in gridworld and cartpole tasks. The curves are presented with standard error bars over 100 independent trials.

coding produces a more robust action selection that is unaffected by the volatility in the spiking activity of a single neuron. To achieve a similar affect, we incorporate population coding by decomposing an agent into a population of agents or equivalently considering a population of networks.

We average the action probabilities across a population of networks and the final action is chosen with a softmax action selection. The individual networks are then updated in an off-policy manner with a positive TD error if the action chosen by the network is same as the final action of the ensemble and a negative TD error otherwise.

## 2.5 Case studies

In this section, we apply our framework to solve two sample RL tasks,[1] a delayed reward task (gridworld) and a continuous control task (cartpole).

**Gridworld**: Consider a $10 \times 10$ gridworld domain with four possible actions (Up, Down, Left, Right) at state. Transition to the terminal state at $(10, 10)$ has a reward of 10 and every other transition has a reward of zero. The 100 states of gridworld are encoded using 3 neurons with a spike train length of 5. The hidden layer has 5 agents each with a spike train length of 3. The stimulus filter $\mathbf{k}$ of a agent is a kernel of 3 parameters which produces the hidden layer spike train responses upon convolution with the spike train stimuli from the previous layer. For simplicity we ignore the other filters. The output layer has 4 agents each corresponding to an action of the domain and the activity of the agent is encoded in a single spike. A population of 10 such networks are concurrently used to select the actions. In Figure 3(a), learning curve from the above network is compared against a tabular actor-critic (AC) and an AC parameterized by neural networks (15 input - 50 hidden - 4 output) and trained with backpropagation. The best hyperparameters for each of the methods are tuned separately. The comparison between the neural network AC and spiking agent AC is not a fair one ( the former has a neural network critic whereas the latter has a tabular critic) but gives a rough sense of learning in the spiking agent AC.

**Cartpole**: In this experiment we consider a simplified spiking agent network where actions of each agent are single spikes rather than spike trains. We train this network to solve the cart pole balancing task Florian (2007) to demonstrate the efficacy of the variance reduction methods employed. The task is to balance the pole for 200 time steps with two possible actions and reward of 1 for each time step that the pole remains balanced. We use 4 input neurons to represent the value of each of the 4 state variables. The hidden layer has 200 agents and the output layer has 2 agents for the two actions. Actions are chosen using a population of 10 such networks. Figure 3(b,c) demonstrates that incorporation of modularity and population coding makes the framework conducive to local learning.

---

[1] The code implementing the spiking agent network can be found at `https://github.com/asneha213/spiking-agent-RL`

We have demonstrated through a couple of toy problems that our framework is conducive to local learning when appropriate variance reduction techniques are incorporated. Application of this learning technique to solve more complex RL tasks might require introduction of advanced variance reduction techniques. However, it is a step towards biologically plausible reinforcement learning.

## 3   Conclusion

In this paper, we extended the concept of a hedonistic neuron Klopf (1982), Seung (2003) by formulating a spiking neuron as an RL agent. A noisy spiking neuron with temporal coding is proved to have more computational power than a sigmoidal neuron Maass (1997) which is yet to be harnessed. A group of such agents organized in a hierarchical and modular fashion interacting with each other in cooperation and competition has a potential for rich representational learning. We have demonstrated one such learning framework which is capable of solving RL tasks thus underscoring the relevance of the neuroscientific principles for the advancement of artificial intelligence.

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
