# OpenReview forum: "Reinforcement learning with a network of spiking agents"
_NeurIPS.cc/2019/Workshop/Neuro_AI — Real Neurons & Hidden Units @ NeurIPS 2019 Poster_

### Official Review · AnonReviewer3 · 2019-09-26
**Nice study of adapting neuroscience principles for solving RL tasks**

**Clarity:** 4

**Comment:**

The authors develop multi-agent learning framework with spiking neurons to solve reinforcement learning tasks. Authors adapt generalized linear model (GLM) as spiking agent and use local learning rules modulated by global reward prediction error to train the network. In addition, authors complement the framework with brain inspired modular architecture and population coding to reduce the variance in learning updates. Authors applied the framework to two RL tasks to demonstrate its potential as viable optimization technique.

The value in this work is that authors adapt brain inspired principles such as spiking neuron, modularity and population coding into their framework and demonstrate each principle contributes to learning in RL tasks. The successful adaptation of neuroscience principles in this work is a good example of how neuroscience can promote a novel framework for AI.

**Category:**

Neuro->AI

**Clarity Comment:**

The paper is easy to follow and overall well written.

**Evaluation:**

4: Very good

**Importance:**

3: Important

**Importance Comment:**

The ideas presented here are novel as they show how neuroscience principles such as modularity and population coding can be adapted to achieve successful learning for RL tasks.

**Intersection:**

4: High

**Intersection Comment:**

The paper is well positioned in the intersection of AI and neuroscience, and shows how knowledge from neuroscience continues to inspire new novel frameworks for AI.

**Rigor Comment:**

The technical information provided is sufficient for following the paper. However I wish the materials under 2.4. were explained in further depth in terms of equations defined in 2.1 - 2.3.

**Technical Rigor:**

3: Convincing

---

### Official Review · AnonReviewer2 · 2019-09-27
**A good illustration of the effect of architecture when learning with global error signals**

**Clarity:** 3

**Category:**

Neuro->AI

**Clarity Comment:**

The writing is generally quite good, though there are a few parts that are either a bit imprecise or hard to parse. e.g.

- I don't understand the sentence that begins at the end of page 2.
- The neuroscience of "modular structures" invoked in section 2.4 is vague.
- The jump from "population coding" to "ensemble model" seems a bit unmotivated.
- I don't understand the first sentence of the cartpole task description.
- The term "computational power", in reference to [Maass, 1997] is vague.

**Evaluation:**

3: Good

**Importance:**

3: Important

**Importance Comment:**

Making the best use possible of global error signals may be very important for solving challenging machine learning tasks with a neurally plausible algorithm.

**Intersection:**

5: Outstanding

**Intersection Comment:**

Learning to accomplish standard tasks in the ML::RL community using global error signals, with neurally inspired variance reduction techniques seems like a good fit for this workshop.

**Rigor Comment:**

The experiments are clear and demonstrate the efficacy of the two proposed variance reduction techniques.

However, I see two technical issues that may limit the scope of this work:

1) It seems the number of timesteps simulated is very low (5 for gridworld, if I'm interpreting "spike train length of 5" correctly), which makes it unclear how the networks described relate to event-driven spiking networks operating in continuous time, since the representation sparsity is so different and the information throughput of the cells in the paper seems limited. For example, using ensembles may have less of an effect on networks with more information throughput per cell. It would be good to compare an ensemble of 10 networks to a single network with 10 times as many cells, and a single network running with 10 times the temporal granularity.

2) The networks tested were very small, and Fig. 3b shows cells struggling to learn from 200 inputs. This makes me unsure how well the proposed approach can scale.

Also, it seems the networks in Fig. 3a may not have finished training.

**Technical Rigor:**

2: Marginally convincing

---

### Official Review · AnonReviewer1 · 2019-09-30
**Nice idea; might not scale**

**Clarity:** 4

**Category:**

Common question to both AI & Neuro

**Clarity Comment:**

-

**Evaluation:**

3: Good

**Importance:**

3: Important

**Importance Comment:**

The idea of considering individual cells as "firing policies" might advance learning in spike-based systems.

**Intersection:**

4: High

**Intersection Comment:**

The presented work is at the intersection of Neuro + AI

**Rigor Comment:**

The idea of learning spike train generation through RL is interesting, however, it is questionable if this method will scale well to larger systems. In particular, as its number of communication partners increases, a neuron has to deal with a more non-stationary environment, making learning in systems of non-trivial size hard.
Thus, it is unlikely that a framework of this type would work in systems with more realistic sizes (i.e. number of neurons on the order of neurons in biological brains.) It would be nice if some results for larger systems (e.g. tens of neurons) could be shown...

**Technical Rigor:**

2: Marginally convincing

---

### Decision · Program_Chairs · 2019-10-02

Accept (Poster)